# GaPP2, a multicentre randomised controlled trial of the efficacy of gabapentin for the management of chronic pelvic pain in women: study protocol

Katy Vincent,[1] Andrew Baranowski,[2] Siladitya Bhattacharya,[3] Judy Birch,[4] Ying Cheong,[5] Roman Cregg,[2] Jane Daniels,[6] Catherine A Hewitt,[6] Gary J Macfarlane,[7] Lee Middleton,[6] Wojciech Szubert,[8] Irene Tracey,[9] Amanda C de C Williams,[10] Andrew W Horne[8]

For numbered affiliations see end of article.

**Correspondence to**
Dr Andrew W Horne;
andrew.horne@ed.ac.uk

## ABSTRACT

**Introduction** Chronic pelvic pain (CPP) affects more than 1 million UK women with associated healthcare costs of £158 million annually. Current evidence supporting interventions when no underlying pathology is identified is very limited and treatment is frequently inadequate. Gabapentin (a GABA analogue) is efficacious and often well tolerated in other chronic pain conditions. We have completed a successful pilot randomised controlled trial Gabapentin for Pelvic Pain 1 (GaPP1) and here describe the protocol for our definitive multicentre trial to assess the efficacy of gabapentin in the management of CPP in women Gabapentin for Pelvic Pain 2 (GaPP2).

**Methods and analysis** We plan to perform a double-blind placebo-controlled randomised multicentre clinical trial, recruiting 300 women with CPP from up to 40 National Health Service hospitals within the UK. After randomisation, women will titrate their medication (gabapentin or placebo) over a 4-week period to a maximum of 2700 mg or placebo equivalent and will then maintain a stable dose for a 12-week period. Response to treatment will be monitored with validated questionnaires and coprimary outcome measures of average and worst pain scores will be employed. The primary objective is to test the hypothesis that treatment with gabapentin has the potential to provide an effective oral treatment to alleviate pain in women with CPP in the absence of any obvious pelvic pathology.

**Ethics and dissemination** Ethical approval has been obtained from the Coventry and Warwick Research Ethics Committee (REC 15/WM/0036). Data will be presented at international conferences and published in peer-reviewed journals. We will make the information obtained from the study available to the public through national bodies and charities.

**Trial registration number** ISRCTN77451762; Pre-results.

## INTRODUCTION

Chronic pelvic pain (CPP) affects more than 1 million women in the UK.[1–3] It is associated with significantly reduced quality of life (QoL),[4 5] a 45% reduction in work productivity and it has been estimated that caring for women with CPP in the UK costs £158 million annually.[6 7] CPP can be associated with underlying pathology such as endometriosis, but in up to 55% of women no obvious cause can be identified at laparoscopy.[6] Management of CPP is difficult when no pathology is identified, as no established gynaecological treatments are available. Due to its effectiveness in other chronic pain conditions, gabapentin (a GABA analogue), is increasingly being prescribed for CPP in both primary and secondary care.[8] However, there is no good quality evidence in CPP specifically on which to base this practice.[9] To our knowledge, there is only one study evaluating the use of gabapentin for CPP, which did not have a placebo arm.[10] This small study in 56 women, compared gabapentin with amitriptyline and showed gabapentin to have greater efficacy at improving pain scores at 12 months. However, efficacy of gabapentin has been proven in other chronic pain conditions. A recent high-quality review showed the number needed to treat to be 5.8 (95% CI 4.3 to 9.0) to achieve at least 50% pain intensity reduction in painful diabetic neuropathy (829 patients); 7.5 (95% CI 5.2 to 14) to achieve at least 50% pain intensity reduction in postherpetic neuralgia (892 patients) and 5.4 (95% CI 2.9 to 31) to achieve at least 30% pain intensity reduction in fibromyalgia (150 patients).[8] Moreover, it is a drug that is very well tolerated: all-cause withdrawal rates are similar to placebo (gabapentin: 20%; placebo: 19%; number of studies: 17; number of participants: 3063).[8]

Given the clinical need for a medical treatment for CPP with no identifiable underlying pathology and the strong evidence supporting the acceptability and efficacy of gabapentin in other chronic pain conditions, we considered that further investigation of gabapentin as a potential treatment for CPP in women was warranted. We hypothesise that treatment of women with CPP in the absence of any obvious pelvic pathology with gabapentin will alleviate pain and improve physical and emotional functioning. We initially performed a successful pilot randomised controlled trial (RCT) (GaPP1).[11 12] Here, we describe the protocol for our definitive multicentre trial to assess the efficacy of gabapentin in the management of CPP in women (GaPP2).

## Objectives
### Primary objective
The primary objective is to test the hypothesis that treatment with gabapentin has the potential to provide an effective oral treatment to alleviate pain in women with CPP in the absence of any obvious pelvic pathology.

### Secondary objective
The secondary objective is to test the hypothesis that treatment with gabapentin has the potential to improve physical and emotional functioning in women with CPP in the absence of any obvious pelvic pathology.

## Outcomes
### Primary outcome
We will employ coprimary outcome measures of average and worst pain scores recorded on a Numerical Rating Scale (NRS). To capture the cyclicity that may occur with CPP, weekly pain scores (on a 0–10 scale) will be recorded during the final 4 weeks of treatment (weeks 13–16 postrandomisation), in the form of: (1) 'average pain this week' and (2) 'worst pain this week'.

The composite 'average' pain score will be taken as the average of the four weekly average pain scores submitted, and the composite 'worst' pain score as the worst of the four weekly worst pain scores submitted.

### Secondary outcomes
► Physical and emotional function and QoL.
► Satisfaction with treatment.
► Patient estimate of whether on active treatment or on placebo group, and confidence in and reasons for estimate.
► Adherence to trial treatments, as reported by the participants.
► Concomitant analgesic use, as reported by the participants.
► Adverse events, as reported by participants (principally those that are serious and detailed in the summary of product characteristics and those that are unexpected).
► General practitioner/hospital consultations, as reported by the participants.
► Time off work and 'presenteeism'.

## METHODS AND ANALYSIS
### Study design
GaPP2 is a double-blind placebo-controlled randomised multicentre clinical trial (figure 1). We will screen women with CPP from up to 40 National Health Service hospitals within the UK. Women will return weekly NRS pain scores to the trials office for 4 weeks after initial consent. Those women meeting the inclusion criteria at the end of these 4 weeks will be randomised. We will randomise 300 women (150 to gabapentin, 150 to placebo). After randomisation and titration, participants will receive treatment with the maximum tolerated dose for 12 weeks. Participants and the healthcare team will be unblinded at the end of their treatment.

### Participants
A total of 300 women with a history of CPP with no obvious pelvic pathology detected at laparoscopy will be recruited to the trial.

### Inclusion criteria
► Women aged between 18 and 50 years.
► CPP (with or without dysmenorrhoea or dyspareunia) of >3-months duration.
► Pain located within the true pelvis or between and below anterior iliac crests.
► No obvious pelvic pathology at laparoscopy (laparoscopy must have taken place at least 2 weeks ago, but no more than 36 months prior to screening).
► Using or willing to use effective contraception if necessary to avoid pregnancy.
► Able to give informed consent.
► For both the worst and average prerandomisation NRS questions, at least three of the four weekly scores returned to the trials office. At least two of the worst pain scores should be ≥4/10. Potential participants who have been on a stable dose of an analgesic, other than gabapentin or pregabalin, for at least 4 weeks prior to screening will be eligible.

### Exclusion criteria
► Dysmenorrhoea alone
► known pelvic pathology:
  – endometriosis (macroscopic lesions)
  – complex or >5 cm ovarian cyst
  – fibroid >3 cm
  – dense adhesions
► current malignancy under treatment
► current use of gabapentin/pregabalin
► taking Gonadotrophin Releasing Hormone (GnRH) agonists and unable/unwilling to stop
► surgery planned in the next 6 months
► history of significant renal impairment
► previous allergic reaction to gabapentin
► breast feeding
► pregnant
► planning pregnancy in next 6 months

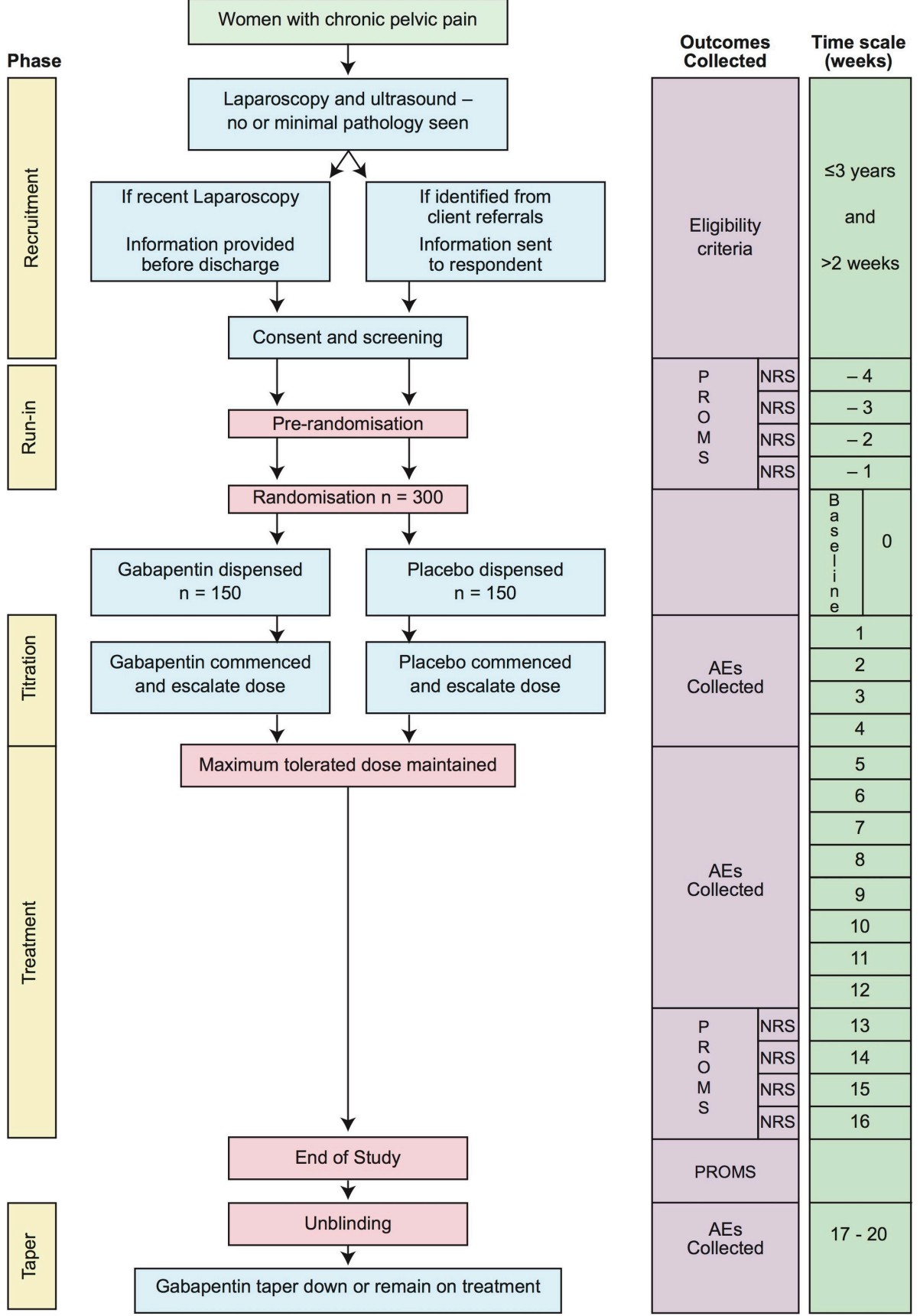

**Figure 1**  Study flow chart. AE, adverse event; NRS, Numerical Rating Scale; PROM, patient-reported outcome measure.

- ► pain suspected to be of gastrointestinal origin (positive Rome III Diagnostic Criteria)
- ► coenrolment in another Clinical Trial of an Investigational Medicinal Product.

## Participant enrolment

Research nurses (dedicated or through the National Institute for Health Research's Clinical Research Network, depending on the site) will be employed for the duration of the study to approach eligible women, provide them with patient information sheets and offer them the opportunity to discuss the trial, and obtain informed consent for screening. Consent will only be taken once the patient has had ample time to read the patient information sheet and had her questions answered.

## Study settings

We will recruit patients from gynaecology outpatient clinics, gynaecology wards and day surgery units and CPP clinics within the UK.

## Intervention and randomisation

Randomisation to gabapentin or placebo will occur once written informed consent has been obtained, final eligibility established from the pain responses provided during the screening phase and baseline questionnaires completed. The Birmingham Clinical Trials Unit (BCTU) will provide third-party web-based randomisation with telephone backup. A minimisation procedure using a computer-based algorithm will be used to avoid chance imbalances in treatment allocation and the following potentially important variables:

1. presence or absence of dysmenorrhoea (a pain score of ≥4/10 will be considered significant);
2. psychological distress measured by the General Health Questionnaire (GHQ) (scored as 0–12 with a cut-off of 0–1 and 2–12 for minimisation);
3. use of sex hormonal treatments (combined oral contraceptive, progestogens, levonorgestrel-releasing intrauterine system (Mirena));
4. centre.

A 'random element' will be included in the minimisation algorithm, so that each patient has a probability (unspecified here) of being randomised to the opposite treatment that they would have otherwise received. Full details of the algorithm used will be stored in a confidential document at BCTU. Both participants and the research team will remain blind to allocation.

## Dose regimen

After randomisation, participants will be allocated a trial treatment pack from the hospital pharmacy containing either gabapentin or placebo oral tablets, both of identical appearance. Participants will start on one capsule (300 mg) daily and will increase by one capsule (300 mg) increments every 3 days until they perceive that they are gaining adequate pain relief, or report side effects (eg, dizziness, somnolence, mood changes, appetite and poor concentration)

that preclude them from further increases, up to a maximum dose of nine capsules (2700 mg), as shown in table 1. The titration phase will last a maximum of 4 weeks. If necessary they will be advised to titrate down to the last tolerated dose with minimal side effects. They will be asked to maintain their best tolerated dose until the end of week 16. Patients will be advised and given written instructions regarding their dosing regimen by a member of the research team. It will be recommended that the drug should be taken in three equally divided doses daily. The same protocol will be used for the placebo. When the participant stops treatment, the dose will be reduced according to a dose reduction chart and written instructions will be given. Patients will be allowed to use other medication (including analgesics, self-medication and alternative treatments, eg, acupuncture) throughout the study period.

## Data collection

### Data storage

All the data generated from the study will be stored in a bespoke database, which will be password protected. All paperwork will be kept in a locked filing cabinet in a locked office. All data will be stored in accordance with the Data Protection Act.

### Screening

A member of the research team will assess the woman for eligibility to enter the screening phase. All data will be recorded on a case report form (CRF) and transferred to a secure database, which will trigger the start of the weekly collection of pain scores.

### Participant log

The clinical research team will keep an anonymised electronic log of women who fulfil the eligibility criteria, women who are invited to participate in the study, women recruited and women who leave the trial early. Reasons for non-recruitment (eg, non-eligibility, refusal to participate and administrative error) will also be recorded. During the course of the study, we will document reasons for withdrawal from the study and lost to follow-up if available.

### Pain scores

Pain NRS will be collected by an automated text messaging system. Two texts will be sent to the women's mobile phone, asking about average and worst pain, respectively, and the woman will be asked to reply to the text message with her pain score, rating it from 0 (for no pain at all) to 10 (being worst pain imaginable). To capture cyclicity, these will be collected weekly during the eligibility phase (weeks −1 to −4) and during the last 4 weeks of the treatment phase (weeks 13–16).

### Treatment diaries

Participants will be provided with a treatment diary at the same time as their medication pack is dispensed.

**Table 1** Dose escalation schedule for GaPP2

| Day in study | Total number of capsules/day (maximum) | Dosing | Maximum daily dose of gabapentin (mg) |
|---|---|---|---|
| 1 | 1 | One capsule night | 300 |
| 2 | 1 | One capsule night | 300 |
| 3 | 1 | One capsule night | 300 |
| 4 | 2 | One capsule two times a day | 600 |
| 5 | 2 | One capsule two times a day | 600 |
| 6 | 2 | One capsule two times a day | 600 |
| 7 | 3 | One capsule three times a day | 900 |
| 8 | 3 | One capsule three times a day | 900 |
| 9 | 3 | One capsule three times a day | 900 |
| 10 | 4 | One capsule two times+two capsules at night | 1200 |
| 11 | 4 | One capsule two times+two capsules at night | 1200 |
| 12 | 4 | One capsule two times+two capsules at night | 1200 |
| 13 | 5 | Two capsules two times+one capsule once | 1500 |
| 14 | 5 | Two capsules two times+one capsule once | 1500 |
| 15 | 5 | Two capsules two times+one capsule once | 1500 |
| 16 | 6 | Two capsules three times a day | 1800 |
| 17 | 6 | Two capsules three times a day | 1800 |
| 18 | 6 | Two capsules three times a day | 1800 |
| 19 | 7 | Two capsules two times+three capsules night | 2100 |
| 20 | 7 | Two capsules two times+three capsules night | 2100 |
| 21 | 7 | Two capsules two times+three capsules night | 2100 |
| 22 | 8 | Three capsules two times+two capsules once | 2400 |
| 23 | 8 | Three capsules two times+two capsules once | 2400 |
| 24 | 8 | Three capsules two times+two capsules once | 2400 |
| 25 | 9 | Three capsules three times a day | 2700 |
| 26 | 9 | Three capsules three times a day | 2700 |
| 27 | 9 | Three capsules three times a day | 2700 |
| 28–112 | | Remain on maximum tolerated dose until the end of week 16 (not exceeding 2700 mg or nine capsules per day). Daily dose should be divided equally into three doses. | |

GaPP2, Gabapentin for Pelvic Pain 2.

The following measures will be completed by the participant daily from day 1 of treatment until week 16:
- ▶ dose of gabapentin taken
- ▶ reason for any change in trial medication dose
- ▶ alternative therapies used
- ▶ any visits to a healthcare professional.

### Questionnaires

A questionnaire will be given to all participants before randomisation but after screening (baseline) and at 16 weeks postrandomisation (see table 2 for full schedule of assessments). This will include the following validated tools:
- ▶ 12-Item Short-Form Health Survey (SF-12): a QoL measure.[13]
- ▶ Brief Pain Inventory (BPI)[14]: a tool to measure pain intensity and interference of pain in a patient's life.
- ▶ Brief Fatigue Inventory (BFI).[15]
- ▶ GHQ[16]: to identify psychological distress.
- ▶ Work and Productivity Activity Impairment Questionnaire (WPAIQ).[17]
- ▶ Pain Catastrophising Scale (PCS).[18]
- ▶ Sexual Activity Questionnaire (SAQ).[19]
- ▶ PainDETECT: to identify a neuropathic component to pain.[20]
- ▶ Pelvic Pain and Urinary/Frequency Patient Symptom Scale (PUF) (at baseline only).

The questionnaire at baseline will include questions to capture the baseline demographic and clinical characteristics of the participants.

**Table 2** Schedule of outcome assessments for GaPP2

| Phase | Run-in | Baseline, randomisation and treatment dispensed | Titration | Treatment | | End of study and unblinding | Taper |
|---|---|---|---|---|---|---|---|
| Duration (weeks) | −4 to −1 | 0 | 1–4 | 5–12 | 13–16 | | 17–19 |
| Weekly worst and average NRS | x x x x | | | x x x | x | | |
| SF-12 | | X | | | | X | |
| BPI | | X | | | | X | |
| PCS | | X | | | | X | |
| SAQ | | X | | | | X | |
| BFI | | X | | | | X | |
| GHQ | | X | | | | X | |
| WPAIQ | | X | | | | X | |
| PainDETECT | | X | | | | X | |
| PUF | | X | | | | | |
| Adverse events | | | X | X | X | X | X |
| Permitted/concomitant medication | X | | X | X | X | | X |
| Adherence or discontinuation | | | X | X | X | | X |

BFI, Brief Fatigue Inventory; BPI, Brief Pain Inventory; GHQ, General Health Questionnaire; NRS, Numerical Rating Scale; PCS, Pain Catastrophising Scale; PUF, Pelvic Pain and Urinary/Frequency Patient Symptom Scale; SAQ, Sexual Activity Questionnaire; SF-12, 12-Item Short-Form Health Survey; WPAIQ, Work and Productivity Activity Impairment Questionnaire.

All questionnaires will be anonymised and completed in private.

### Adverse events

Participants will collect information about adverse events in their treatment diaries. However, they will be instructed to contact the clinical research team at any time after consenting to join the trial if they have an event that requires hospitalisation or an event that results in persistent or significant disability or incapacity. Any serious adverse events (SAEs) that occur after joining the trial will be reported in detail in the participant's medical notes and followed up until resolution of the event. The assessment of seriousness, causality and expectedness will be conducted assuming that the participant received gabapentin, with the blinding not broken. All SAEs will be reported to the Academic and Clinical Central Office for Research and Development (ACCORD) Research Governance (http://www.accord.ed.ac.uk) and Quality Assurance Office based at the University of Edinburgh immediately or within 24 hours. ACCORD will onward report all SAEs to BCTU within 7 days.

### Termination of study

Participants will be unblinded at the end of the study and if taking gabapentin will have the option to continue on treatment or will be tapered off treatment. Participants who have been on placebo will be given the choice to start on gabapentin, which will be prescribed by their clinician.

Participants will be given an emergency contact card to carry while participating in the study. The blinding code will only be broken in emergency situations for reasons of patient safety, where knowledge of the treatment administered is necessary for the treatment of a SAE. Participants whose randomisation codes are broken will cease treatment with the study drug, but will continue to be followed up. Participants may discontinue from the trial at any time at their own request, or they may be withdrawn at any time at the discretion of the research team for safety, behavioural or administrative reasons. Data collection is envisaged to be complete in September 2018.

### Sample size

We have based our sample size on being able to detect a minimally important difference (MID) in NRS scores with high levels of power. Studies have shown the MID in this population to be around 1 point on a 0–10 scale.[21] Our pilot study showed worst and average pain scores to have SDs between 2.0 and 2.5. If the SD is at the lower end of these estimates, 86 patients in each group (172 in total) would be required to have 90% power (P=0.05) to detect a difference of 1 point. If the SD is at the higher end, we could detect the same difference with 80% power (P=0.05) with 100 patients in each group. We have assumed the latter SD (2.5) to be conservative. To account for any increase in the risk of type I error that may be associated with having coprimary outcome measures, we have

applied a Bonferroni correction (alpha reduced to 0.025 from 0.05), which increases the sample size to 120 per group. Furthermore, to account for an expected average 20% lost to follow up, we will randomise 150 per group, 300 patients in total.

## Proposed analyses

Data analysis will be by intention to treat. Every attempt will be made to gather data on all women randomised, irrespective of compliance with the treatment protocol. Appropriate baseline characteristics, split by treatment group, will be presented for each outcome. Point estimates, 95% CIs and P values from two-sided tests will be reported.

### Primary analysis

We will use a linear regression model to estimate differences in worst and average NRS scores between the two treatment groups, including baseline score and the minimisation variables as covariates. The P value from the associated $\chi^2$ test will be produced and used to determine statistical significance. A Bonferroni correction will be applied as there are two primary outcomes. Further analysis using a repeated measures (multilevel) model will also be performed incorporating all eight recorded scores.

### Secondary analysis

Data from the other continuous measures (SF-12, BPI, PCS, SAQ, WPAIQ, BFI, PainDETECT and GHQ) will be analysed in a similar manner to the primary measure. Other outcome measures (use of permitted analgesic medication, satisfaction) will be analysed using standard methods (tests for trend, absolute/relative risks). Further analysis on pain scores will include an examination of the proportion of women that have a 30% and a 50% reduction in average and worst score from baseline as the outcome. A log-binomial regression model will be used here to generate adjusted relative risks. Subgroup analyses will be limited to the same variables that were used as minimisation variables. Tests for statistical heterogeneity (eg, by including treatment group by subgroup interaction parameter in the linear regression model) will be performed prior to any examination of effect estimate within subgroups. In addition, we will investigate up to nine clinical variables measured at baseline to determine whether they correlate with response to treatment. These will include the minimisation variables (the presence of dysmenorrhoea/psychological distress/current use of hormonal treatment) along with measures of intensity and of nature of pain (eg, PainDETECT), number of functional systems involved (as a measure of organ specific versus generalised pelvic pain syndrome) and PUF score.

### Missing data and sensitivity analyses

Every attempt will be used to collect full follow-up data on all women. In particular, participants will continue to be followed up even after protocol treatment violation.

It is thus anticipated that missing data will be minimal. Patients with completely missing primary outcome data or with only one of four pain scores recorded will not be included in the primary analysis. Secondary sensitivity analyses will be performed to investigate the impact of missing data for the primary outcome: this will include a worst score assumption. We will also simulate missing responses using a multiple imputation approach.

### Research governance

We shall adopt the standard approach used for monitoring RCTs and have a Trial Steering Committee (TSC) of at least four independent members, including pain specialist, a gynaecologist, trial methodologist and a Patient and Public Involvement (PPI) representative. There will also be a Data Monitoring Committee (DMC) comprising three independent members (a pain specialist, a gynaecologist and a statistician with extensive trial experience) who will review interim analyses. The terms of reference and charter for this DMC will be guided by the DAMOCLES project, and we anticipate the DMC and TSC will meet biannually.

## DISCUSSION

CPP is a major public health issue for women throughout the developed world.[2] As with other chronic pain conditions, it is associated with a marked reduction in QoL and significant financial costs for the woman, her family and society as a whole.[4 5] When CPP is associated with underlying pathology such as endometriosis, therapies targeting the pathology can be initiated. However, in more than 50% of women, no underlying cause is identified.[6] For these women, not only is it difficult to comprehend and come to terms with how there can be no associated pathology,[22] there are also no available evidence-based treatments to consider.

The efficacy of a number of pharmacological and interventional therapies has been investigated for other chronic pain syndromes. There is increasing evidence that women with CPP demonstrate central changes similar to those associated with other forms of chronic pain[23 24] and thus it is likely that such therapies would also be effective for CPP. Moreover, recent work demonstrates a neuropathic component in a significant proportion of women with CPP,[25] further supporting the investigation of drugs currently recommended for neuropathic pain[26] in women with CPP. The multicentre placebo-controlled RCT described here aims to contribute to the evidence base by assessing the efficacy of gabapentin in women with CPP with no underlying pathology.[9] This trial is designed in line with the Initiative on Methods, Measurement, and Pain Assessment in Clinical Trials (IMMPACT) recommendations for the design of trials in chronic pain conditions[21 27 28] and builds on a successful pilot study.[11 12] Women with CPP were surveyed to identify whether reduction in average or worst pain was most important to them. As there was no clear consensus (average 43.4%, worst 56.6%) coprimary outcomes of average and

worst pain scores have been chosen. We envisage the findings being of relevance to both primary and secondary care clinicians managing women with CPP.

**Author affiliations**
[1]Nuffield Department of Obstetrics and Gynaecology, University of Oxford, Oxford, UK
[2]Pain Management Centre, The National Hospital for Neurology and Neurosurgery, London, UK
[3]Institute of Applied Health Sciences, School of Medicine, Medical Sciences and Nutrition, University of Aberdeen, Aberdeen, UK
[4]Pelvic Pain Support Network, Poole, UK
[5]Faculty of Medicine, University of Southampton, Southampton, UK
[6]Birmingham Clinical Trials Unit, University of Birmingham, Birmingham, UK
[7]Epidemiology Group, School of Medicine, Medical Sciences and Nutrition, University of Aberdeen, Aberdeen, UK
[8]MRC Centre for Reproductive Health, Queen's Medical Research Institute, Edinburgh, UK
[9]Nuffield Department of Clinical Neurosciences, University of Oxford, Oxford, UK
[10]Research Department of Clinical, Educational and Health Psychology, University College London, London, UK

**Acknowledgements** We gratefully acknowledge the other investigators that contributed to the design of the RCT: Jayna Holroyd-Leduc, Danielle Southern, Ward Flemons, Maeve O'Beirne, Michael Hill, Alan Forster and Deborah E White.

**Contributors** KV: research, contribution of original material, drafting, editing and approval of final manuscript; AWH, SB, RC, JD, LM, WS: research, contribution of original material, editing and approval of final manuscript; AB, IT, JB, YC, GJM, ACCW: contribution of original material, editing and approval of final manuscript.

**Funding** GaPP2 is funded by a grant from the NIHR/MRC Efficacy and Mechanism Evaluation (EME) Programme, Reference: 13/52/04.

**Competing interests** KV has received research funding from Pfizer Pharmaceuticals and Bayer HealthCare, honoraria from Bayer HealthCare and consultancy fees from Grünenthal GmbH. AWH receives grant funding from the NIHR and the Medical Research Council. AB has no competing issues as no direct funding from industry. AB is currently President of the British Pain Society. JB declares no competing interests. SB receives grant funding from NIHR and Chief Scientist Office Scotland. He has been a speaker at a number of conferences that have received funding from the pharmaceutical industry. His clinical colleagues receive industry support for travel and for Departmental Seminars. ACCW has received a consulting fee from Astellas. GJM is Chief Investigator of the British Society for Rheumatology (BSR) Biologics Register in Ankylosing Spondylitis. This is funded by the BSR who receive funds from Pfizer, AbbVie and UCB. YC receives grant funding from NIHR. RC has secured research funding from Allergan and is a member of Allergan's scientific advisory panel. He also receives grant funding from the NIHR via UCL Biomedical Research Centre. IT is supported by grants from NIHR Oxford Biomedical Research Centre, Medical Research Council of Great Britain and Northern Ireland, the Wellcome Trust.

**Patient consent** Obtained.

**Ethics approval** Ethics approval has been obtained from the Coventry and Warwick Research Ethics Committee (REC 15/WM/0036).

**Provenance and peer review** Not commissioned; externally peer reviewed.

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
