## [Reviewer comments · BMJ Open]

ARTICLE DETAILS

TITLE (PROVISIONAL)	GaPP2: A multi-centre randomised controlled trial of the efficacy of gabapentin for the management of chronic pelvic pain in women: study protocol
AUTHORS	Vincent, Katy; Baranowski, Andrew; Bhattacharya, Siladitya; Birch, Judy; Cheong, Ying; Cregg, Roman; Daniels, Jane; Hewitt, Catherine; Macfarlane, Gary; Middleton, Lee; Szubert, Wojciech; Tracey, Irene; Williams, Amanda; Horne, Andrew

VERSION 1 – REVIEW

REVIEWER	Simon Haroutounian, PhD Washington University School of Medicine St Louis, MO, USA No Competing Interest
REVIEW RETURNED	31-Jan-2017

GENERAL COMMENTS	1. The objectives define safe, effective and convenient oral tx...; however, the outcome of convenience is not well defined and methods for its quantification are not provided. I suggest it is either removed from objectives or information is provided on its assessment, analysis etc.2. Introduction - 1st sentence. The statement of CPP comparison to the prevalence of other conditions is not referenced. Asthma is estimated to affect >5 million patients in the UK, and back pain affects >6.5 million, supported by various NHS publications (vs >1 million stated further in the sentence for CPP). Please provide specific numbers and references to your statement or modify the sentence.3. Page 7 L 12-14. the sentence referring to "average pain score .. as the average of i and ii.." is unclear and confusing. please rephrase.4. In table 2 there is a description of saliva sample, with no justification/mentioning in the text.5. Page 11 lines 40-42. All other medications will be allowed? including the initiation of new analgesics? How would that affect outcomes? What about medications such as antacids that could affect oral bioavailability of gabapentin? What about chronic analgesics that were discontinued during the study period? The capture (and interpretation) of other medications use requires more details.6. Page 11 line 51. Typo "in an in a".
--

	7. Page 12 line 29. Pain sores: Why not capture pain scores weekly during weeks 1-13 to look at the dynamics of pain? it is valuable information, it would be a pity not to collect it. 8. Page 8: participants and study team will be unblinded at the end of the treatment. What processes to you have in place this does not affect your data analyses - e.g. assigning likelihood that a certain adverse effect is related to the study medication. 9. Page 9. Exclusion criteria. Bladder pathologies are not mentioned - what about conditions such as recurrent UTI that result in pelvic pain? What about musculoskeletal pathologies (e.g. prior orthopedic pelvis surgeries) which could result in CPP? 10. Interstitial Cystitis is one of the common reasons for chronic pelvic pain. the study is focused primarily on Gynecological aspects, but IC is not mentioned in the introduction, inclusion, exclusion etc. It would be important to specify if the study plans to include or exclude these patients.
--	---

REVIEWER	Harsha Shanthanna McMaster University, Canada
REVIEW RETURNED	12-May-2017

GENERAL COMMENTS	The study addresses an important question, and a potentially valuable treatment. In view of their pilot study publication and its differences in the methodology and reporting to this study protocol, as well as some important methodological aspects that were unclear, I think the protocol manuscript would need substantial changes to consider a revised version for publication. 1. Funding: Please expand what the funding from NIHR/MRC grant is? What is the value, and whether this is completely non-industry funded? 2. There seems to be several changes compared to your pilot study: treatment and follow up- up to 3 months instead of 6 months; use of HADS vs. Distress scale; other outcome measures, including a single pain efficacy measures vs. average and worse (co-primary). It is important to highlight why these changes were considered or done? 3. You seem to indicate that CPP has a significant element of central sensitization, similar to other chronic pain conditions. That is possibly true. In your introduction, to substantiate the efficacy of gabapentinoids you refer to studies in diabetic neuropathy and postherpetic neuralgia-both of which clearly fall into definite neuropathic pain diagnosis, based on the latest neuropathic diagnostic criteria. While, it is true that gabapentinoids have shown good efficacy in clear neuropathic pain conditions, it is important to note that changes in central sensitization [hyperalgesia or allodynia] do not represent neuropathic pain, but characterise central changes [Woolf CJ. Central sensitization: Implications for the diagnosis and treatment of pain. Pain. 2011;152(3 Suppl):S2-15]. In fact, more and more research emphasizes the need for a multimodal approach [Nijs J, Malfliet A, Ickmans K, Baert I, Meeus M. Treatment of central sensitization in patients with 'unexplained' chronic pain: an update.
--

Expert Opin

Pharmacother. 2014 Aug;15(12):1671-83]. Moreover, gabapentin has shown mixed or no efficacy in many non-neuropathic chronic pain conditions such as chronic low back pain or whiplash pain, both of which are observed to have a predominant central sensitization component.

Specific Questions:

1. Primary Objective:

Primary objective is usually reflected by the primary outcome, which in this case is pain relief. The primary objective stated is "to test the hypothesis that treatment with gabapentin has the potential to provide a SAFE, EFFECTIVE, and CONVENIENT oral treatment". An effective outcome (decreased pain relief) need not necessarily be safe and convenient, as this study is not powered to capture the convenience and safety of gabapentin.

2. Why do you want to consider CO-PRIMARY outcomes?

What is the important of choosing both average and worse pain scores?

What if one of them is significantly better and other is not (for example average pain score is better and worse scores are not?- how do you plan to interpret this? Would you say use gabapentin only if the worse pain score is >4, but need not use if the average pain score >4?

Your pilot study mentions only VAS of 0-10, why was this altered for the final study?

3. Selection Criteria:

1. How did you establish the range of 'no more than 36 weeks and at least 2 weeks' for a negative diagnostic laparoscopy? What if a patient had a negative diagnostic laparoscopy 4 years ago?

2. In page 6, last paragraph, you mention that, to capture the cyclicity that may occur with CPP, weekly pain scores----. However, you are excluding patients who may have non-cyclical pain {2nd line-inclusion criteria]: you need worry about cyclical pain?

3. Current malignancy under treatment: what malignancies are covered and why? What if it's a skin malignancy?

4. What would you do if a patient has another source of chronic pain (such as back pain or neck pain), which is severe than CPP?

5. How are you controlling for patients who would be put on other analgesics (opioids, antidepressants)-for other sources of pain?

6. Why are you excluding a patient who has surgery planned in the next 6 months? Again what surgery are you referring to?

7. What does previous reaction to Gabapentin mean? Allergy?

8. What is CTIMP?

4. Recruitment:

Why are you aiming to recruit these patients in day surgery units? Your inclusion needs at least 2 weeks after a negative diagnostic laparoscopy. You are not expected to approach a patient soon after their surgery?

5. Allocation concealment: How will the drugs be packaged? Who does it? And who delivers them to the patient?

6. Secondary Outcomes:

Why do you want to capture patient estimate of whether on active or placebo treatment, does this inference make your study more or less valid? [Sackett DL. Turning a blind eye: why we don't test for blindness at the end of our trials. BMJ. 2004 May 8;328(7448):1136].

7. Gabapentin has been shown to predominantly effect neuropathic pain. Are you using PainDETECT to do a subgroup analysis, or even have a threshold of PainDETECT to consider using in the minimization procedure for randomization? Please see my comments on Neuropathic pain vs. Central sensitization pain.

8. Sample Size:

The MID of 1 in a 0-10 point scale, cited by IMMPACT is wrong. IMMPACT recommendations are based on an MID of 2 points or 30% reduction [Farrar. Pain. 2001 Nov;94(2):149-58]. This is the same article which IMMPACT uses to substantiate the MID of 2 points.

9. Analysis:

How would you define compliance-what is the threshold? Is it complete treatment, or treatment for at least 2 months? Or all patients taking treatment in the last 4 weeks (because that captures your primary outcome).

You may want to consider per protocol analysis (sensitivity analysis) on patients who are compliant according to your threshold. You seem to have performed this in the pilot study.

10. Primary Analysis:

It is not clear if you are looking at the difference between baseline and final scores or only final pain scores (average of the last 4 weeks) for both average and worse pain scores?

11. Is there any DSMB (drug safety monitoring board?)

12. Is there any interim assessment?

13. Will you be continuing to follow up patients beyond 3 months?

14. OFF LABEL use:

Are you aware of that there can be misuse of gabapentin and it typically happens in off-label uses?

<https://www.ncbi.nlm.nih.gov/pubmed/14664664>

<https://www.ncbi.nlm.nih.gov/pubmed/24760436>

Since Gabapentin is being prescribed for an off-label use, would you not need the Medicines and Healthcare products Regulatory Agency (MHRA), approval for a clinical study? Has this been obtained?

VERSION 1 – AUTHOR RESPONSE

Reviewer 1

Comment and Responses:

1. The objectives define safe, effective and convenient oral tx...; however, the outcome of convenience is not well defined and methods for its quantification are not provided. I suggest it is either removed from objectives or information is provided on its assessment, analysis etc.

RESPONSE: We agree with the reviewer and have removed this from the protocol.

2. Introduction - 1st sentence. The statement of CPP comparison to the prevalence of other conditions is not referenced. Asthma is estimated to affect >5 million patients in the UK, and back pain affects >6.5 million, supported by various NHS publications (vs >1 million stated further in the sentence for CPP). Please provide specific numbers and references to your statement or modify the sentence.

RESPONSE: The figures quoted came from a legitimate, but old, source and refer to the prevalence of these conditions in women. However, we have modified this sentence.

3. Page 7 L 12-14. the sentence referring to "average pain score .. as the average of i and ii.." is unclear and confusing. please rephrase.

RESPONSE: This has been rephrased.

4. In table 2 there is a description of saliva sample, with no justification/mentioning in the text.

RESPONSE: This has been removed. It refers to a pharmacogenomic substudy we had planned.

5. Page 11 lines 40-42. All other medications will be allowed? including the initiation of new analgesics? How would that affect outcomes? What about medications such as antacids that could affect oral bioavailability of gabapentin? What about chronic analgesics that were discontinued during the study period? The capture (and interpretation) of other medications use requires more details.

RESPONSE: This is an important point. However, having shared our experience of data collection with our panel of pain medicine experts and trialists, it was deemed that it would be impossible to enforce or monitor in this trial. We will not be documenting whether women have taken antacids.

6. Page 11 line 51. Typo "in an in a".

RESPONSE: This has been corrected.

7. Page 12 line 29. Pain scores: Why not capture pain scores weekly during weeks 1-13 to look at the dynamics of pain? it is valuable information, it would be a pity not to collect it.

RESPONSE: We collected this information in our pilot study but the feedback from participants at the end of the trial was that this was too onerous. We have therefore compromised by only collecting pain scores over two 4-week periods.

8. Page 8: participants and study team will be unblinded at the end of the treatment. What processes do you have in place this does not affect your data analyses - e.g. assigning likelihood that a certain adverse effect is related to the study medication.

RESPONSE: All clinical report forms will be completed before unblinding. The assessment of the seriousness, causality and expectedness of adverse events will be conducted by both the local and chief investigator without being unblinded and under the assumption that the participant is taking gabapentin. A line to clarify this has been added to page 14.

9. Page 9. Exclusion criteria. Bladder pathologies are not mentioned - what about conditions such as recurrent UTI that result in pelvic pain? What about musculoskeletal pathologies (e.g. prior orthopaedic pelvis surgeries) which could result in CPP?

RESPONSE: We appreciate that recurrent UTIs and prior orthopaedic pelvis surgeries could lead to chronic pelvic pain in women. However, after careful discussion with pain medicine clinicians, we decided to limit our exclusion criteria to maximize external validity (generalisability) of the trial.

10. Interstitial Cystitis is one of the common reasons for chronic pelvic pain. the study is focused primarily on Gynecological aspects, but IC is not mentioned in the introduction, inclusion, exclusion etc. It would be important to specify if the study plans to include or exclude these patients.

RESPONSE: We have not excluded women with symptoms of bladder pain syndrome ('interstitial cystitis'). Instead, we are asking women to report daytime and/or night-time urinary frequency (measured by the Pelvic Pain and Urinary/Frequency [PUF] Patient Symptom Scale) (Urol. 2002 60:573–578) at screening (see Page 14) and we will add these variables to our exploratory analysis of what is predictive of a reduction in pain (see addition to 'Secondary analysis' Page 17).

Reviewer: 2

Comments and Responses:

1. Funding: Please expand what the funding from NIHR/MRC grant is? What is the value, and whether this is completely non-industry funded?

RESPONSE: The NIHR/MRC have funded the trial to the value of £1,208,722 and we receive no industry support.

2. There seems to be several changes compared to your pilot study: treatment and follow up- up to 3 months instead of 6 months; use of HADS vs. Distress scale; other outcome measures, including a single pain efficacy measures vs. average and worse (co-primary). It is important to highlight why these changes were considered or done?

RESPONSE: FOLLOW-UP: A lower than expected proportion of patients followed up to six months (53%) in our pilot alerted us to the need to improve retention in this trial. Following discussion with experts in pain medicine regarding the trial medication and our patient representatives, we therefore decided to reduce the follow-up to three months in an attempt to minimize losses. Co-primary outcome and other outcome measures: When we submitted our trial application for consideration of funding, the referees (nine international experts) and NIHR/MRC board requested that we consider co-primary outcomes, despite the findings from our pilot study. We therefore conducted a rapid survey of women with chronic pelvic pain, via the Pelvic Pain Support Network home website and their Health Unlocked website, asking whether average or worst pain was the most important outcome.

212 women replied, with a nearly equal split between average and worst pain being the most important, supporting the proposal for a co-primary outcome. The decision to include the other outcome measures was based on their performance in the pilot, discussion with experts and feedback from the referees/board.

3. You seem to indicate that CPP has a significant element of central sensitization, similar to other chronic pain conditions. That is possibly true. In your introduction, to substantiate the efficacy of gabapentinoids you refer to studies in diabetic neuropathy and postherpetic neuralgia-both of which clearly fall into definite neuropathic pain diagnosis, based on the latest neuropathic diagnostic criteria. While, it is true that gabapentinoids have shown good efficacy in clear neuropathic pain conditions, it is important to note that changes in central sensitization [hyperalgesia or allodynia] do not represent neuropathic pain, but characterise central changes [Woolf CJ. Central sensitization: Implications for the diagnosis and treatment of pain. *Pain*. 2011;152(3 Suppl):S2- 15]. In fact, more and more research emphasizes the need for a multimodal approach [Nijs J, Malfliet A, Ickmans K, Baert I, Meeus M. Treatment of central sensitization in patients with 'unexplained' chronic pain: an update. *Expert Opin Pharmacother*. 2014 Aug;15(12):1671-83]. Moreover, gabapentin has shown mixed or no efficacy in many non- neuropathic chronic pain conditions such as chronic low back pain or whiplash pain, both of which are observed to have a predominant central sensitization component.

RESPONSE: We are grateful to the reviewer for this helpful comment.

Specific Questions:

1. Primary Objective:

Primary objective is usually reflected by the primary outcome, which in this case is pain relief. The primary objective stated is "to test the hypothesis that treatment with gabapentin has the potential to provide a SAFE, EFFECTIVE, and CONVENIENT oral treatment". An effective outcome (decreased pain relief) need not necessarily be safe and convenient, as this study is not powered to capture the convenience and safety of gabapentin.

RESPONSE: We have removed 'convenient' from the primary outcome. However, we have kept the word 'safe' because participants will collect information about adverse events in their treatment diaries. This information will be analysed at the end of the trial.

2. Why do you want to consider CO-PRIMARY outcomes? What is the important of choosing both average and worse pain scores? What if one of them is significantly better and other is not (for example average pain score is better and worse scores are not?-how do you plan to interpret this? Would you say use gabapentin only if the worse pain score is >4, but need not use if the average pain score >4? Your pilot study mentions only one VAS of 0-10, why was this altered for the final study?

RESPONSE: See response to reviewer 2, question 2. Both outcomes will be reported and interpreted separately. If one outcome is found to be significantly better and the other is not we will state that evidence of effect was found with respect to worst pain but not average pain (or vice-versa). To account for any possibility of increased risk of false positive finding this has been accounted for in the sample size calculation.

3. Selection Criteria:

a. How did you establish the range of 'no more than 36 months and at least 2 weeks' for a negative diagnostic laparoscopy? What if a patient had a negative diagnostic laparoscopy 4 years ago?

RESPONSE: The lower end of the range was to ensure the baseline pain measurement scores were not influenced by any post-operative pain from the diagnostic laparoscopy. The upper end of the timeframe was a clinical consensus regarding how long following a negative laparoscopy would it be reasonable to conclude the pain should be viewed as idiopathic, and when would the majority of clinicians suggest a second look laparoscopy in women with persistent pelvic pain. However, we accept that this is an arbitrary timeframe due to the lack of good evidence regarding the interval incidence of pathologies.

b. In page 6, last paragraph, you mention that, to capture the cyclicity that may occur with CPP, weekly pain scores----. However, you are excluding patients who may have non-cyclical pain {2nd line-inclusion criteria]: you need worry about cyclical pain?

RESPONSE: We apologise if this is confusing in the protocol. We are recruiting women with chronic pelvic pain (with or without dysmenorrhoea) but we are excluding women who present with dysmenorrhoea only. We have amended the inclusion criteria and added a statement to the exclusion criteria to make this clearer. We are capturing pain scores over four weeks because we acknowledge that chronic pelvic pain in women often has a cyclical component (e.g. exacerbated around menstruation and/or ovulation).

c. Current malignancy under treatment: what malignancies are covered and why? What if it's a skin malignancy?

RESPONSE: A woman under treatment for any malignant condition (including a skin malignancy) will be excluded.

d. What would you do if a patient has another source of chronic pain (such as back pain or neck pain), which is severe than CPP?

RESPONSE: They are not excluded from the trial. We are looking to achieve a degree of pragmatism and offer gabapentin to women who represent the full spectrum of the population to which it might ultimately be applied (and this could include women with other pain conditions). The primary outcome specifically asks about pelvic pain.

e. How are you controlling for patients who would be put on other analgesics (opioids, antidepressants)-for other sources of pain?

RESPONSE: The use of concomitant medications will be collected using treatment diaries completed by the participants as necessary throughout the course of the trial, and this is a secondary outcome (see Page 13 and 7, respectively).

f. Why are you excluding a patient who has surgery planned in the next 6 months? Again what surgery are you referring to?

RESPONSE: We are excluding women who have planned surgery because of the likelihood that they will require additional analgesia intra- and post-operatively with potential impact on our outcomes.

g. What does previous reaction to Gabapentin mean? Allergy?

RESPONSE: Yes – this has been amended to make it clearer (see Page 9).

h. What is CTIMP?

RESPONSE: A CTIMP is a 'Clinical Trial of an Investigational Medicinal Product'. We have stated this term in full in the protocol.

4. Recruitment:

Why are you aiming to recruit these patients in day surgery units? Your inclusion needs at least 2 weeks after a negative diagnostic laparoscopy. You are not expected to approach a patient soon after their surgery?

RESPONSE: The women may be approached for consideration of the trial in day surgery units but they will not be randomised from there.

5. Allocation concealment: How will the drugs be packaged? Who does it? And who delivers them to the patient?

RESPONSE: Sharp Clinical Services (<http://www.sharpservices.com>) will be manufacturing and packaging the trial medication. This is being done on a commercial basis according to Good Manufacturing Practice and distributed to hospital pharmacies, who will dispense the number treatment packs. We have clarified this on page 11.

6. Secondary Outcomes:

Why do you want to capture patient estimate of whether on active or placebo treatment, does this inference make your study more or less valid? [Sackett DL. Turning a blind eye: why we don't test for blindness at the end of our trials. *BMJ*. 2004 May 8;328(7448):1136].

RESPONSE: In practice, perfect blinding is impossible to ensure or verify post hoc. It is a current accepted standard to ask the trial participants to guess their treatment allocation. These responses can be used to compute the extent of blinding in the trial in the form of a blinding index. If the estimated extent of blinding exceeds a particular threshold the trial is deemed sufficiently blinded; otherwise, the validity of the study can be questioned. There is debate about how to statistically use this information (see Arandjelovic 2012) but we will in the first instance, test the deviation from an expected 50:50 ratio of guessed allocations, which would occur if the trial drug was truly blinded.

7. Gabapentin has been shown to predominantly effect neuropathic pain. Are you using PainDETECT to do a subgroup analysis, or even have a threshold of PainDETECT to consider using in the minimization procedure for randomization? Please see my comments on Neuropathic pain vs. Central sensitization pain.

RESPONSE: We have not minimised the randomisation by pain type categorisation using the PainDETECT questionnaire. If we were to do a post-hoc sub-group analysis, this would not be powered to detect a difference, and so any observed would be considered hypothesis generating and have to be interpreted very cautiously.

8. Sample Size: The MID of 1 in a 0-10 point scale, cited by IMMPACT is wrong. IMMPACT recommendations are based on an MID of 2 points or 30% reduction [Farrar. *Pain*. 2001 Nov;94(2):149-58]. This is the same article which IMMPACT uses to substantiate the MID of 2 points.

RESPONSE: The paper by Dworkin et al summarises three attempts to derive a MID in pain studies (Farrar 2001, Salaffi 2004, Hanley 2006) and describes a difference of 1 point on a 0-10 NRS (or 15-20% decrease) as representing the minimally important difference and 2 points (or 30-36% decrease) as "much better" or a meaningful difference. We believe even a minimal improvement would be worthwhile detecting and hence have used this as our target treatment effect. This also means we are being conservative – a 2-point difference would results in a very small clinical trial.

9. Analysis: How would you define compliance-what is the threshold? Is it complete treatment, or treatment for at least 2 months? Or all patients taking treatment in the last 4 weeks (because that captures your primary outcome). You may want to consider per protocol analysis (sensitivity analysis) on patients who are compliant according to your threshold. You seem to have performed this in the pilot study.

RESPONSE: Compliance will be monitored by self-reported treatment diaries which are returned to the trial team at Visit 5 (week 16). We will define compliance in the following categories: $\geq 75\%$ good compliance; 50-75% reasonable compliance; $< 50\%$ poor compliance. The 'per protocol' cohort is defined as only those participants with good compliance to allocation. A full statistical analysis plan has been drafted and approved by DMC and TSC committees which covers a more comprehensive view of compliance and alternative analysis populations. This will ultimately be made available with the trial publication.

10. Primary Analysis: It is not clear if you are looking at the difference between baseline and final scores or only final pain scores (average of the last 4 weeks) for both average and worse pain scores?

RESPONSE: Both average and worst pain scores are adjusted for baseline score. We feel this is clear in the analysis section.

11. Is there any DSMB (drug safety monitoring board?)

RESPONSE: We shall adopt the standard approach used for monitoring randomised controlled trials and have a Trial Steering Committee (TSC) of at least four independent members, including pain specialist, a gynaecologist, trial methodologist and a PPI representative. There will also be a Data Monitoring Committee (DMC = 'DSMB') comprising three independent members (ideally a pain specialist, a gynaecologist and a statistician with extensive trial experience). The terms of reference and charter for this DMC will be guided by the DAMOCLES project, and we anticipate the DMC and TSC will meet biannually. This statement has been added to the protocol.

12. Is there any interim assessment?

RESPONSE: During recruitment, a report containing an interim analysis of the primary and available secondary outcome measures will be provided in strict confidence to the DMC, but will not be published. We have added a section called "Research Governance" on page 18.

13. Will you be continuing to follow up patients beyond 3 months?

RESPONSE: No.

14. OFF LABEL use: Are you aware of that there can be misuse of gabapentin and it typically happens in off-label uses? <https://www.ncbi.nlm.nih.gov/pubmed/14664664>
<https://www.ncbi.nlm.nih.gov/pubmed/24760436>. Since Gabapentin is being prescribed for an off-label use, would you not need the Medicines and Healthcare products Regulatory Agency (MHRA), approval for a clinical study? Has this been obtained?

RESPONSE: We are aware of the potential for gabapentin misuse and we have obtained MHRA approval for our trial.

We hope look forward to hearing from you.

VERSION 2 – REVIEW

REVIEWER	Simon Haroutounian, PhD Washington University School of Medicine, St Louis, MO, USA No Competing Interest
REVIEW RETURNED	19-Sep-2017

GENERAL COMMENTS	The authors have addressed the majority of concerns. The recruitment to the trial has begun, so that at this point it is not feasible to address few of the suggestions for improving the study design. I would encourage the authors to submit their future protocols for publication in a peer-reviewed journals prior to initiating enrollment; otherwise there is not much value in the peer review process.
--

REVIEWER	Harsha Shanthanna McMaster University Canada
REVIEW RETURNED	19-Sep-2017

GENERAL COMMENTS	I just have the following objections or questions with the revised manuscript. 1. You state that the primary objective is still to assess the “safety and effectiveness of gabapentin”. I do not also agree with the statement that they are safe drugs, overall for chronic pain patients. I want to bring it to your notice some recent publications in high impact literature that bring attention to their adverse effects.1. http://www.nejm.org/doi/full/10.1056/NEJMp1704633#t=article2. Shanthanna H, Gilron I, Rajarathinam M, AlAmri R, Kamath S, Thabane L, et al. (2017) Benefits and safety of gabapentinoids in chronic low back pain: A systematic review and meta-analysis of randomized controlled trials. PLoS Med 14(8): e1002369.3. Mathieson S, Maher CG, McLachlan AJ, Latimer J, Koes BW, Hancock MJ, Harris I, Day RO, Billot L, Pik J, Jan S, Lin CC. Trial of Pregabalin for Acute and Chronic Sciatica. N Engl J Med. 2017 Mar 23;376(12):1111-1120.4. Wise J. Gabapentinoids should not be used for chronic low back pain, meta-analysis concludes. BMJ. 2017 Aug 16;358:j3870.Recommendation or Suggestion: In view of this, one cannot simply assume that it would be both effective and safe. In your trial, as you rightly have noted in page 7 (secondary outcomes), adverse effects are secondary outcomes, hence it should be considered as a secondary objective along with other secondary outcomes. Your trial is not powered to assess its safety. Hence, I would request that the primary objective be modified as “assessing the effectiveness of gabapentin”, without including the safety perspective, both in abstract and full text. Also, in page 14, please take out the statement [Gabapentin is generally well tolerated in the management of other chronic pain conditions, and serious adverse events are not anticipated] or add that there is potential for adverse effects.2. Please check the references for accuracy. For example, the sentence “Studies have shown the MID in this population to be around 1 point on a 0-10 scale”, is supported by citation 20. However, it seems that it should have been 21 (Dworkin et al.)
--

	3. Thanks for clarifying the primary outcomes. Although your trial is set up to find the average/worst pain score differences between the group, it is suggested that investigators also provide the proportion of patients benefitted, either by 20% or 30% or both. Average or group scores don't support much clinical decision making. Could you consider including them as one of the secondary outcomes and report it in the final analysis? 4. Testing whether you had true blinding or not? I agree that this can be tested, but the question is whether you would report your findings as more biased if analysis based on a deviation of more than 50%, shows that the blinding was not effective? Again, it could be due to chance, because the trial is not powered? Best wishes, Harsha Shanthanna
--	--

VERSION 2 – AUTHOR RESPONSE

Reviewer: 1

We are pleased that Reviewer 1 is happy with our response to their comments.

Reviewer: 2

Comments and Responses:

1. I would request that the primary objective be modified as “assessing the effectiveness of gabapentin”, without including the safety perspective, both in abstract and full text. Also, in page 14, please take out the statement [Gabapentin is generally well tolerated in the management of other chronic pain conditions, and serious adverse events are not anticipated] or add that there is potential for adverse effects.

RESPONSE: We have modified the primary outcome and removed this statement.

2. Please check the references for accuracy. For example, the sentence “Studies have shown the MID in this population to be around 1 point on a 0-10 scale”, is supported by citation 20. However, it seems that it should have been 21 (Dworkin et al.)

RESPONSE: We have checked the references and corrected this error.

3. Thanks for clarifying the primary outcomes. Although your trial is set up to find the average/worst pain score differences between the group, it is suggested that investigators also provide the proportion of patients benefitted, either by 20% or 30% or both. Average or group scores don't support much clinical decision making. Could you consider including them as one of the secondary outcomes and report it in the final analysis?

RESPONSE: This is a very helpful comment and we will consider it in our final analysis.

VERSION 3 – REVIEW

REVIEWER	Harsha Shanthanna McMaster University Canada
REVIEW RETURNED	25-Sep-2017
GENERAL COMMENTS	I appreciate the effort taken. Thanks